# Dielectric Barrier Discharge for Solid Food Applications

**DOI:** 10.3390/nu14214653

**Published:** 2022-11-03

**Authors:** María Fernanda Figueroa-Pinochet, María José Castro-Alija, Brijesh Kumar Tiwari, José María Jiménez, María López-Vallecillo, María José Cao, Irene Albertos

**Affiliations:** 1Faculty of Health Sciences, Universidad Católica de Ávila (UCAV), 05005 Ávila, Spain; 2Recognized Research Group: Assessment and Multidisciplinary Intervention in Health Care and Sustainable Lifestyles, University of Valladolid, 47003 Valladolid, Spain; 3Faculty of Nursing, University of Valladolid, 47003 Valladolid, Spain; 4Teagasc Food Research Centre, D15 DY05 Dublin, Ireland

**Keywords:** atmospheric cold plasma (ACP), dielectric barrier discharge (DBD), food decontamination

## Abstract

Atmospheric cold plasma (ACP) is a non-thermal technology whose ability to inactivate pathogenic microorganisms gives it great potential for use in the food industry as an alternative to traditional thermal methods. Multiple investigations have been reviewed in which the cold plasma is generated through a dielectric barrier discharge (DBD) type reactor, using the atmosphere of the food packaging as the working gas. The results are grouped into meats, fruits and vegetables, dairy and lastly cereals. Microbial decontamination is due to the action of the reactive species generated, which diffuse into the treated food. In some cases, the treatment has a negative impact on the quality. Before industrializing its use, alterations in colour, flavour and lipid oxidation, among others, must be reduced. Furthermore, scaling discharges up to larger regions without compromising the plasma homogeneity is still a significant difficulty. The combination of DBD with other non-thermal technologies (ultrasound, chemical compounds, magnetic field) improved both the safety and the quality of food products. DBD efficacy depends on both technological parameters (input power, gas composition and treatment time) and food intrinsic properties (surface roughness, moisture content and chemistry).

## 1. Introduction

The World Health Organization states that microbiologically originating foodborne illnesses are a major threat to public health. This microbial growth is likewise the leading cause of food spoilage [1]. In the past, the means to preserve food for safe consumption involved the use of heat treatments. These unfortunately included impairing chemicals and organoleptic product properties. As a result, the demand for more natural, minimally processed products has been increased by consumers. In particular, industrialized countries have increased their interest in products that use minimal processing. Minimal processing is defined as products whose nutritional content and freshness has not been significantly changed so as increase its safety and preservation. The more commonly used preservation techniques in the food industry are: high-pressure processing, high intensity pulsed electric field, pulsed light and irradiation [2,3,4,5,6,7,8].

Another non-thermal technology is cold plasma. Industrial cold plasma equipment has not been implemented in the food industry yet. Cold plasma is composed by gas molecules, positive or negative ions and free radicals. The gas temperature is close to ambient temperature and it is obtained at atmospheric or reduced pressure (vacuum). Cold plasma does not present a local thermodynamic equilibrium [9,10]. Although several review articles on cold plasma have been published discussing antimicrobial efficacy [9,11,12,13] they analyse the atmospheric cold plasma in a general way. However, there are no available reviews focused on dielectric barrier discharge (DBD) efficacy. There are several sources for producing cold plasma such as corona discharges, plasma jet and glow discharge [14].

Dielectric barrier discharges (DBD) are generated when high voltage is applied across the electrodes. These discharges generate energetic electrons that dissociate oxygen molecules by direct impact. This single O atom combines with oxygen molecules (O_2_) to form ozone gas [15].

The purpose of this review conducted was to provide the potential of the application of DBD in different solid food groups. It included the mechanism of action in microbial inactivation. The combination of DBD with other technologies has been discussed (hybrid technologies and hurdle technologies). A hurdle technology is a minimal processing technology that exploits synergistic interactions between DBD and other preservation treatment. It concluded with a discussion of the main challenges and industrial application.

## 2. Methods

A literature review has been carried out on the inactivation of pathogenic microorganisms and spoilage microflora in food by the application of DBD. The Preferred Reporting Items for Systematic Reviews and Meta-Analyses (PRISMA) guidance was followed.

The databases consulted were mainly Web of Science and Pubmed. The search was carried out using Boolean operators: AND (mainly) and OR. The following terms were used Health Sciences Descriptors (DeCS): Atmospheric cold plasma, and Medical Subject Heading (MeSH): Atmospheric cold plasma. Both DeCS and MeSH allow the use of common terminology in multiple languages.

The search was conducted using the following keywords: “cold plasma AND food”, “cold plasma AND microorganism”, “cold plasma AND bacteria”, “DBD AND food”, “DBD AND microorganism”, “ACP AND food decontamination”, “cold plasma AND meat”, “cold plasma AND fruits”, “cold plasma AND dairy”, “cold plasma AND cereals”.

Articles with the following characteristics were excluded from this research: articles older than 5 years, articles in a language other than English, articles not relevant or belonging to journals not indexed in JCR, articles in which work was done with gases or gas mixtures other than ambient air and articles in which only liquid foods were used.

The process used to identify and select articles is shown in Figure 1. When we started the research in the WOS database, we entered the words “atmospheric cold plasma” obtaining 4.798 articles as a result, but the articles were not related to the main topic of this study, thus we started to filter the information. While reviewing some of the first articles listed, we decided to focus the research and include to the previous phrase the words: “AND food”, and the research in all databases included in this site (Web of Science Core Collection, Current Contents Connect, Derwent Innovation Index, KCL Korean journal Databases, Medline, Russian Science Citation, Scielo Citation index). The results were reduced to 834 articles. Replacing the word “food” with “microorganism” yielded 885 articles (last 5 years). The information obtained from this research corresponded to the first section of the results of this work, for the second section we used the phrase: “cold plasma AND bacteria”, obtaining 717 results. As this number of articles was not enough, a serch by food type was performed: “DBD AND food” with 275 results (with filter: published last 5 years), “DBD AND microorganism” with 320 results as well, “ACP AND food decontamination” with 48 results, “cold plasma AND meat” with 207 articles as results and “cold plasma AND fruits” with 353 results. It was necessary to filter out the articles that used DBD-type reactors and also ambient air as a gas and the only way to obtain this information was by reading the abstract of the study and/or reading the description of the reactor used in it. For this reason, it was necessary to filter by food (“cold plasma AND lettuce”, “cold plasma AND chicken” etc.).

## 3. Mechanism of Action

There are several theories to explain the mechanism of action of cold plasma at atmospheric pressure in microbial inactivation. Reactive oxygen and nitrogen species generated in DBD can lead to cell membrane rupture (Figure 2), or to the oxidation of lipids, amino acids and nucleic acids with reactive oxygen and nitrogen species. These reactions lead to microbial death or injury. On the other hand, alteration of microbial DNA by UV photons generated by the plasma [16] create electrostatic forces from the accumulation of charged particles from the plasma, resulting in damage to the cell membrane of the micro-organism, leading to cell death [17].

Gram-positive and gram-negative bacteria respond differently to plasma treatment. In gram-negative bacteria, cell disruption is generated resulting in leakage of intracellular components such as potassium, nucleic acids and proteins [18,19]. In contrast, in gram-positive bacteria there is damage to intracellular components and no cell leakage [20]. This is because gram-positive bacteria have a thick layer of murein (peptidoglycan) as the outer layer of the cell, unlike gram-negative bacteria, which have a thin murein layer, and are therefore less resistant to cold plasma technology [19].

In addition to the reactive species O_3_ and NO_2_, the peroxynitrite anion (ONOO^-^), formed by the reaction between NO and O_2_ radicals generated in the plasma, can also contribute to bacterial inactivation. UV photons inhibit DNA replication with the formation of thymine dimers. In doing so, they directly damage the genetic material [12]. The wavelength range between 200 and 300 nm causes dimerization of thymine bases in DNA strands and disrupts the replication of bacteria [19]. OH radicals generated in plasma lead to the degradation of membrane lipids through a chain of oxidation reactions, resulting in the disintegration of unsaturated lipids into lipid peroxides, which can produce non-radical species, by means of a chain reaction, which can be accelerated by the O_3_ ion. The reactive species generated in plasma also react with amino acids, causing structural changes in proteins [12]. Larger cellular proteins are the first to be degraded. The mechanism attributed to this degradation is the destruction of hydrogen and sulphur bonds by reagents present in the plasma. This results in changes in the primary, secondary and tertiary structure of the protein, leading to a decrease in the enzymatic activity of the cell [18].

On the other hand, the reactive species OH, O, O_3_ and H_2_O_2_ have been related to the breaking of structural bonds in the peptidoglycan component of the cell wall. The bonds mentioned can be C-O and C-N bonds, which would result in the destruction of the cell wall of the microorganism [12].

## 4. Application of DBD in Food for Decontamination

This section presents the results and discussion, ordered by food group. The groups included are meats, fruits and vegetables, dairy products and cereals.

### 4.1. Meat and Fish

The application of DBD in meat has been extensively studied. This group includes research on poultry, pork, beef, lamb and fish. The results have been summarized in Table 1. It has been demonstrated how DBD treatment is able to reduce bacterial counts in both spoilage microflora and pathogenic bacteria inoculated in the laboratory.

Lee et al. [21] applied DBD cold plasma at 22, 23 and 24 kV power for 2, 2.5 and 3 min to packaged chicken breast cubes. The chicken breast cubes were inoculated with a cocktail of *Salmonella* (*S*. *enteritidis*, *S*. *Montevideo* and *S*. *Typhimurium*) and Tulane virus, as a surrogate for human Norovirus. In addition, the spoilage microorganisms were quantified. This treatment resulted in microbial inactivation of naturally occurring mesophilic bacteria, *Salmonella enteritidis* and Tulane virus, at 0.7 ± 0.2 log CFU/cube, 1.4 ± 0.1 logs CFU/cube and 1.1 ± 0.2 logs CFU/cube, respectively. Treatment time has been shown to exert a significant effect on decreasing the population of mesophiles, psychotropic and enterobacteria. The treatment voltage of 24 kV for 3 min resulted in the highest inactivation of *Salmonella* (1.45 ± 0.05 log CFU/cube). Tulane virus was reduced after treatment to 1.08 ± 0.15 log CFU/cube.

Another research group working with packaged chicken breasts is Zhuang et al. [22], who inoculated the samples with approximately 5 CFU/mL with *Salmonella* and *Campylobacter*. In addition, they also studied spoilage bacteria such as psychrophilic bacteria. The packaged samples were treated with 70 kV for 0, 60, 180 and 300 s, then stored at 4 °C for 5 days. The bacterial population in DBD plasma treated chicken breasts decreased significantly (*p* ˂ 0.05) according to the treatment time, the 60 s treatment results in more than 0.5, 0.7 and 0.4 log reductions in psychrophiles, *Campylobacter* and *Salmonella*, respectively. Extending the treatment time beyond 60 s results in no further microbial reductions. Increasing the treatment time to 180 s significantly reduces the psychrophile population by 0.6 logs. Increasing the treatment time again caused no further microbial reductions in the psychrophile population in chicken breasts. This study concludes that according to their data the DBD system in packages can effectively inhibit microbial growth and reduce food pathogens in air-packed chicken breasts. Their results showed a change in the appearance of DBD-treated chicken breasts, as post-treatment samples looked paler. Another important conclusion of this work is that the treatment time does not influence the antimicrobial effectiveness in the case of food pathogens at 70 kV, but for spoilage flora the treatment time can affect the effectiveness of the antimicrobial system in the package.

Patange et al. [23] studied one of the main spoilage microorganisms of raw lamb meat, the bacterium *Brochothrix thermosphacta* and its inactivation by DBD. The treatment of 80 kV power for 30 s was applied on a sample of *Brochothrix thermosphacta* in PBS. On the other hand, the shelf life after plasma treatment was investigated in chops packed in modified atmospheres (MAP) in 30% CO_2_ and 70% O_2_. The result was the reduction of B. thermosphacta populations in PBS (phosphate buffered saline) to undetectable levels. Treatment applied to packaged lamb chops for 5 min resulted in reductions of 2 log cycles. Higher inactivation efficiency was reported with increasing voltage and time applied to planktonic cells. This would be due to the elevation of reactive species, which is associated with a higher bactericidal effect.

In order to study the efficacy of DBD against *B*. *thermosphacta* biofilm, treatment was also carried out to inactivate it with 80 kV of power, for 60, 120 and 300 s. Biofilm-associated cells were significantly reduced (*p* ˂ 0.05) from 8 log CFU mL^−1^ to 4.6 log CFU mL^−1^, after 60 s of DBD treatment. Extending the time to 120 s, a further reduction in bacterial populations was achieved. However, extending to 300 s had no significant effect, indicating a significant retention of metabolic activity. It has been reported that high concentrations of organic compounds in the medium have protective effects for the microorganisms and against the bactericidal effect of plasma, which is why plasma treatment has much faster effects in PBS (phosphate buffered saline) compared to the effects in lamb chops. In addition to this, the results are attributed to the complexity of the surface matrix of the meat and the ability of these bacteria to grow in challenging environmental conditions. The results of this research indicate that DBD cold plasma has the potential to be an alternative process for the safe decontamination of meat.

The research by Han et al. [24] worked with beef packaged with three types of gas mixture: gas 1: 30 % CO_2_ + 70 % N_2_, gas 2: air and gas 3: 30 % CO_2_ + 70 % O_2_. The DBD treatment applied was high voltage (80 kV). *E*. *coli*, *L*. *monocytogenes* and *A*. *aureus* bacteria were undetectable in PBS after 60 s at 80 kV treatment in air. This study worked with cultured and inoculated biofilms and planktonic cells of these bacteria. The nutritional components of the meat exerted a protective role towards the pathogens, so it was found that this characteristic could inhibit the penetration of ROS into the bacterial cell, which leads to a decrease in the efficacy of the cold plasma treatment. The 300 s exposure to DBD plasma produced a maximum reduction of 1.5 log, using a high oxygen atmosphere, while the use of air and high nitrogen atmospheres produced a lower antimicrobial efficacy. Another important conclusion of this research is that maintaining the refrigerated storage temperature for 24 h at 4 °C will maintain the microbial reductions achieved by the cold plasma treatment.

The study by Huang et al. [25] worked with pork loin packed in modified atmospheres (MAP) with three different types of gases: gas 1: 20% MAP (20% O_2_ + 60% N_2_ + 20% CO_2_), gas 2: 40% MAP (40% O_2_ + 40% N_2_ + 20% CO_2_), gas 3: 60% MAP (60% O_2_, 20% N_2_ + 20% CO_2_). The pork loins were treated with high voltage DBD, 85 kV for 60 s and stored at 4 °C for 12 days. The results of this study yielded consistently lower total aerobic counts in the pork loin in the treated group than in the untreated group, with these differences becoming prominent on days 4 and 8. Microbial log reductions increased with increasing oxygen concentrations. This research highlights that the type of reactive species and their concentrations depend on the working gas, so gas composition can affect microbial inactivation rates. Due to the colour and oxidation changes caused by DBD treatment, it is recommended to improve the method.

The research by Albertos et al. [26] studied fresh mackerel fillets treated with DBD plasma at voltages of 70 and 80 kV, for 1, 3 and 5 min. There was no significant reduction (*p* > 0.05) in total mesophilic aerobic counts, but psychrotrophic bacteria, lactic acid bacteria (LAB) and pseudomonas counts were significantly reduced (*p* < 0.05) due to DBD. The reduction of total aerobic bacteria was more dependent on the plasma exposure time than on the applied potency. However, the greatest reduction was obtained with the highest voltage and time (80 kV, 5 min). After 24 h of treatment, spoilage bacteria were significantly reduced, however a significant increase in lipid oxidation was observed. The time and voltage applied in the treatment had an important influence on its antimicrobial effect. This antimicrobial treatment is recommended for fish; however, it should be improved in terms of lipid oxidation.

Another study with fish is by Mohamed et al. [27], which focuses on tilapia quality characteristics and endogenous enzyme activity, but also counts spoilage bacteria through total viable cell count (TVC). The results of this study revealed that the duration and voltage of therapy had a significant impact on enzyme activities, TVC and other measured parameters, and thus the ideal treatment parameters for tilapia were found to be 60 kV for 4 min. An important conclusion of this study is the increase in the shelf life of DBD-treated tilapia to 10 days, while untreated tilapia had a shelf life of only 4 days. From this research it can be deduced that DBD can be used as a positive technology to preserve product quality and prolong shelf life.

Albertos et al. [28] studied the effects of DBD on quality and spoilage microorganisms in packaged herring fillets (*Clupea Harengus*). The results of this study showed a remarkable decrease of spoilage microorganisms (*p* ˂ 0.05) compared to control samples. DBD plasma applied under 70 kV conditions for 5 min (minimum voltage applied in this study) appears to be a potential technology to preserve or prolong the shelf life of fresh fish fillets, while altering important quality characteristics such as oxidation and colour as little as possible.

Chen et al. [29] investigated the efficacy of DBD to improve the quality of mackerel during storage by studying its chemical, microbial and sensory characteristics. In this study, DBD plasma treatment was performed, and samples were subsequently stored in a refrigerator at 4 ± 1 °C for 16 days. The optimum power and exposure time of this treatment were 60 kV and 60 s, respectively. Under these conditions, microbial inactivation was effective, without altering the chemical composition of the treated product. When the treatment time was extended to 75 s, the cell population was no longer significantly reduced. This is probably due to the saturation of the charged particles. The results indicated that the cold plasma treatment extended the shelf life of the mackerel to 14 days, while the untreated samples only reached 6 days. Lipid oxidation and myofibrillar protein degradation were also delayed by the PFA treatment.

Finally, Choi et al. [30], who also worked with fish products, in this case with dried monkfish, which was inoculated with 8–9 log CFU/mL of Staphylococcus aureus and Bacillus cereus. Reductions in bacterial counts in DBD-treated products at 1, 5, 10, 20 and 30 min ranged from 0.10 to 1.03 log CFU/g and 0.14 to 1.06 log CFU/g, respectively. The visual appearance, colour, flavour, and overall acceptability of a dried monkfish treated with DBD plasma were not significantly different between the untreated and plasma-treated dried monkfish. However, when treated with DBD plasma for more than 20 min, the texture value decreased significantly. This study indicates that treatment with DBD plasma for 30 min resulted in more than 90% reduction of *S*. *aureus* and *B*. *cereus*.

### 4.2. Fruits and Vegetables

Table 2 summarizes studies with fresh fruits and vegetables. Undoubtedly this food group has been the most studied so far. Tappi et al. [31] worked with freshly cut melon. It was treated with DBD plasma for 30 and 60 min at 15 kV. Qualitative, metabolic and microbiological indices were analysed for 4 days at 10 °C. The qualitative parameters were slightly affected by the treatment. Microbiological parameters resulted in a significant increase in product shelf life, due to a delay in post-treatment growth of mesophilic and psychrotrophic flora. A 3.4 and 2 log reduction, respectively, was observed for mesophilic and lactic acid bacteria, whose cell loads were below the detection limit at 60 min after treatment. On the other hand, cell load reductions not exceeding 1 log CFU/g were recorded in the psychrotrophs. Compared to the control group, treated melons spoiled after four days, whereas melons in the control group spoilt after 2.5–3 days.

On the other hand, Pasquali et al. [32] have studied cold plasma on red chicory and its decontaminant effectiveness and impact on vegetable quality, as an alternative to traditional chlorine washing, on a large scale. Chicory leaves were inoculated with *Escherichia coli* O157:H7 (gram negative) and *Listeria monocytogenes* (gram positive). The former was significantly reduced after 15 min of cold plasma treatment (1.35 log MPN/cm^2^). However, to achieve a significant reduction of *L*. *monocytogenes* it was necessary to prolong the plasma treatment to 30 min, reaching 2.2 log CFU/cm^2^ in the results. The results after 3 days of storage at 4 °C confirmed the decontamination effect of this treatment, as the log reduction of *L*. *monocytogenes* was maintained throughout the storage period, with no regrowth. After treatment, no quality defects were observed. However, after 1 day of storage, a significant visual quality was decreased compared to the control.

Manhot et al. [33] studied the efficacy of high voltage treatment on the microflora of carrots. The voltages applied were 60, 80 and 100 kV, for 1, 2, 3, 4 and 5 min. A reduction of approximately 2 log CFU/g in total aerobic mesophilic population, and yeast/mould, was achieved in carrots treated at 100 kV for 5 min. The appropriate voltage to increase the shelf life of carrot is between 80 and 100 kV, the ideal time between 4 and 5 min. In this way the altering microflora decreases, but the quality of the carrot is maintained, with small pH, colour, texture and total carotenoids alterations.

Kim et al. [34], studied the effect of multiple packaging parameters on the inactivation of Salmonella inoculated on a mix of vegetables (grape tomato, romaine lettuce, red cabbage and carrot) and indigenous aerobic mesophilic bacteria, yeasts and moulds present on these vegetables. Films made of low-density polyethylene (LDPE), polypropylene and polyethylene terephthalate (PET) were used in the packaging. Three forms of packaging were produced for the experiment: a PET clamshell container (rigid), a rigid container consisting of the support frame and LDPE film and a flexible bag. For the gas composition CO_2_ was the control gas, set at 10%, while the nitrogen concentration varied between 0% and 90%, and the rest was oxygen. In this study, a significantly higher level of *Salmonella* inactivation was observed in grape tomatoes in the flexible bag made of the nylon/LDPE composite than those in the LDPE bag. These differences can be partly explained by the dielectric conductivity of the different packaging materials, which may induce different levels of discharge and thus different amounts of reactive species in the cold plasma. Non-significant differences were found in romaine lettuce, red cabbage and carrot slices depending on packaging material for any of the microorganisms studied. There may be a limitation in contact between reactive species and microorganisms as a result of the complicated surface structure. During treatment without shaking, grape tomatoes in the rigid containers showed a more effective inactivation of *Salmonella* of ~0.4 log CFU/tomato than those in the flexible bags: this may be due to the narrower space between the food and the container in the bag containers than in the rigid containers, which prevents uniform distribution of the reactive species in the container and leads to inefficient contact between the reactive species and the food. When shaking was applied during treatment, there was no significant difference in the level of *Salmonella* inactivation according to the shape of the container (*p* > 0.05). Hence, agitation increases the contact between reactive species and microorganisms on sample surfaces. The results of this study did not indicate any significant difference in microbial inactivation by DBD applied to mixed vegetables packed in MAP with varying concentrations of oxygen or nitrogen. This may be due to the various reactive species produced by oxygen, nitrogen and air, which are known to be effective for microbial inactivation. Inactivation of *Salmonella* by DBD in grape tomatoes increased from 1.3 ± 0.4 log CFU/tomato to 2.0 ± 0.2 log CFU/tomato (*p* < 0.05) when secondary packaging was applied. The application of secondary packaging can improve the inactivation efficiency, as a new material is added as a dielectric barrier and the thickness of the entire packaging is altered. In addition, inactivation of *Salmonella* on the surfaces of grape tomatoes was more effective when the packaged foods were close to the electrodes, thus receiving direct treatment, rather than remote treatment. Therefore, this study suggests that optimal packaging conditions that showed the most effective microbial inactivation include the use of low-density polyethylene as the packaging material, rigid packaging as the packaging shape, a minimum distance between the sample and the electrode and secondary packaging. In addition, the use of agitation is an effective way to improve microbial inactivation.

In the case of strawberries, multiple authors have experimented with this fruit. Ziuzina et al. [35] investigated the efficacy of DBD high voltage cold plasma in static and continuous mode for decontamination and quality retention of strawberries and spinach inoculated with *E*. *coli* and *L*. *innocua*. In selecting products, structural and surface characteristics were taken into consideration: strawberries have a spherical geometry and convolutions on their surfaces, whereas spinach leaves have a lamellar profile. Results achieved bacterial reductions of 2.0 and 2.2 log CFU/mL for *E. coli* and reductions of 1.3 and 1.7 log CFU/mL for *L*. *innocua*, respectively. The continuous treatment was effective against *L*. *innocua* inoculated strawberries, with a reduction of 3.8 log CFU/mL. There were no significant differences in colour, firmness, pH or total soluble solids between the control and the cold plasma treatment group. DBD in static mode significantly reduced strawberry mesophilic bacteria populations by 1.8 log CFU/sample, while only a 0.8 log reduction of mesophiles was achieved in continuous mode. Both treatment modes reduced strawberry yeast and mould populations by an average of 1.2 log CFU/sample. The generation of cold plasma discharges inside food packaging can achieve rapid bactericidal effects through the retention of reactive species during product storage. However, according to the current results, different products require the application of specific treatment parameters. Greater reductions for *E*. *coli* on strawberries and spinach were achieved with the static treatment application, while for *L*. *innocua* inoculated strawberries, reductions were greater with the continuous treatment.

Misra et al. [15] also worked with strawberries, which were treated in-package with 60 kV DBD pulsed at 50 Hz. Strawberry microflora (aerobic mesophilic bacteria, yeasts and moulds) was reduced by 2 log after 5 min of treatment within 24 h of DBD plasma application. The consequences of DBD related to product quality, such as colour and firmness, were negligible. The behaviour of the plasma, its action on microorganisms and the resulting changes in food quality are determined by the plasma chemistry and the resulting dynamics. This involves a large number of different reactive species. DBD plasma in packaging can reduce the spoilage flora present in strawberries, without inducing significant respiratory stress, or negatively affecting the colour and firmness of the treated food.

Rana et al. [36] used DBD on microbial inactivation, physicochemical properties and shelf life of strawberries with prolonged storage in the container at ambient (25 °C) and refrigerated (4 °C) temperatures. The shelf life of DBD-treated strawberries was extended up to 5 days at 25 °C and 9 days at 4 °C in sealed packaging. However, untreated packaged strawberries degraded within 2 days. Treatment with DBD for 15 min resulted in a 2 log reduction in spoilage microorganisms. The 15 min treatment at 60 kV was observed to increase total phenolic content and antioxidant activity. The rest of the parameters studied (pH, moisture and total soluble solids) were not affected by the treatment. Therefore, the 15 min DBD treatment with 5 days storage in the container resulted in an increase in the shelf life and functional quality of the strawberry.

Ziuzina et al. [37] inoculated *Escherichia coli*, *Salmonella Typhimurium* and *Listeria monocytogenes* on cherry tomatoes and strawberries. They were packaged in air and submitted to DBD at 70 kV for 10, 60 and 120 s. *Salmonella*, *E*. *coli* and *L*. *monocytogenes* populations in tomato were at undetectable levels after treatment. In strawberries, higher treatment was required. A possible explanation is the fruit’s surface irregularities. They can act as a niche in which bacteria can grow, providing a physiological barrier or protection from cold plasma treatments. As a result, strawberries were less effective than tomatoes at killing gram-negative bacteria due to this factor. To achieve optimal effectiveness in decontamination by DBD the following factors should be considered: type of products, their inherent surface characteristics, the type of bacteria, the strength and nature of their adhesion and the diffusion capacity of the plasma species.

Wu et al. [38] investigated the effects of DBD treatment on cherries during storage. After treatment, a significant reduction (non-specific bacterial colonies) was observed in the treated groups. Among the treated groups, treatment at 60 kV can inactivate most microorganisms. The use of higher voltages does not offer an excessive advantage in terms of inactivation. Similar results were obtained when different treatment times were considered. The treatment of 100 s at 60 kV was effective in terms of microbial inactivation. The cold plasma treatment effectively removed microorganisms from the surface of the cherries.

Min et al. [39] inoculated *Escherichia coli* O157: H7 on romaine lettuce leaves and evaluated its altering microflora. The treatment conditions were 34.8 kV power 1.1 kHz, for 5 min. DBD treatment reduced *E*. *coli* O157: H7 and total aerobic microorganism counts by an amount similar to 1 log CFU/g. Reductions in the number of *E*. *coli* O157: H7, total mesophilic aerobes and yeasts and moulds during storage were 0.8 to 1.5, 0.7 to 1.9 and 0.9 to 1.7 log CFU/g, respectively. Treatment had no significant effect on colour, CO_2_ formation, weight and surface morphology of lettuce during storage.

Min et al. [40] also inoculated *E*. *coli* O157:H7, *Salmonella*, *L*. *monocytogenes* (6 log CFU/g lettuce) and Tulane virus (2 log CFU/g lettuce) on romaine lettuce leaves treated with DBD plasma at 34.8 kV for 5 min. The microbial reductions obtained were as follows: *E*. *coli* O157:H7, *Salmonella*, *L*. *monocytogenes* and Tulane Virus by 1.1 ± 0.4, 0.4 ± 0.3, 1.0 ± 0.5 log CFU/g and 13 ± 0.1 log PFU/g, respectively.

Zhang et al. [41] showed that DBD treatment at 65 kV for 1 min was effective in reducing spoilage growth of aerobic mesophilic bacteria, yeasts and moulds in pears. This treated fruit also maintained its organoleptic properties and other quality attributes.

Giannoglou et al. [42] treated ready to eat packaged rocket salad with cold plasma. They studied alterations in quality and improvements in shelf life. After DBD treatment, reductions of 0.57 to 1.02 log CFU/g for total microbial load were achieved after 5 and 20 min treatment, respectively. The optimal DBD time to achieve sufficient reduction in microbial load, while maintaining colour and texture, was 10 min.

Min et al. [43] inoculated *Salmonella* on grape tomatoes and treated with DBD at 35 kV for 3 min. The treatment inactivated *Salmonella*. Other properties such as colour or firmness of the tomatoes were not affected. The distribution of the product also played an important role, being more effective in a one layer configuration than double layer.

Kilonzo-Nthenge et al. [44] inoculated packaged Golden Delicious apples with *Salmonella* and *Escherichia coli* and treated at 35 mm spacing at 200 W for 30, 60, 120, 180 and 240 s. In all treatment times, inactivation of the inoculated microorganisms was achieved but higher times of exposure (180 and 240 s) was more effective.

Dong and Yang [45] applied cold plasma treatment at 36 V power for 10 min in blueberries. The number of indigenous bacteria and fungi (yeast/mould) decreased by 93% and 25.8%, respectively. This would occur due to increased DNA damage and guanine oxidation.

### 4.3. Dairy Products

Dairy products must be subjected to thermal processing before marketing, thus avoiding microbiological contamination, toxic substances and off-flavours. Heat sterilisation causes non-enzymatic browning, loss of vitamins and volatile compounds [48]. Table 3 synthesizes experiments performed with DBD plasma in milk products as an alternative to thermal treatment.

Wan et al. [49] inoculated *L*. *innocua* on fresh cheese (QFC) and cheese model (CM) treated with direct and indirect DBD at 100 kV for 5 min. The results with direct treatment were reductions of 3.5 and 1.6 log CFU/g for CM and QFC, respectively. Direct plasma treatment was more effective in inactivating *Listeria innocua* in CM and QFC. The microstructure played an important role in the efficacy of DBD plasma treatment. The roughness of the surface allowed the attachment of bacterial cells to the surface and provided protective sites for them.

The study by Huang et al. [50] studied the efficacy of antimicrobial treatment with DBD applied to cheese inoculated with *Escherichia coli* and *Listeria innocua*. The inactivation increased with higher input power (30, 50, 70 W) and plasma exposure time (0, 1, 3, 5, 7 min). In this research, the distance between electrodes was also studied as a processing parameter. It showed higher levels of microbial inactivation with decreasing distance (2, 1.5, 1 cm). In addition, the types of microorganisms affected the efficacy of DBD plasma.

### 4.4. Cereals

Water, air, dust, soil, insects, birds and rodent faeces can cause microbial contamination of grains. Environmental conditions contribute to this contamination and its persistence. DBD plasma treatment of cereal grains is very attractive, since the chemical method usually used can generate toxic residues in the food. In addition, DBD plasma has been shown to degrade pesticides commonly used in cereal cultivation [51]. Table 4 summarizes research on cereals, such as rice and flour, treated with DBD plasma and ambient air.

Kang et al. [52] have conducted research on Korean packaged rice cakes (KRC), inoculated with *Salmonella* and treated with 25 kV DBD plasma for 1 and 3 min. Subsequently, the effectiveness in eradicating both *Salmonella* and spoilage flora was studied. Treatment with DBD in the container significantly reduced the amount of viable yeast/mould and *Salmonella* to 1.7 ± 0.3 and 3.9 ± 0.3 log CFU/g, respectively. The fungi’s chitin-containing cell membrane is protected from microbial inactivation. Colour and firmness were not affected by DBD treatment. Analysis indicated that cold plasma treatment damaged the *Salmonella* cell membrane due to lipid peroxidation. Further studies such as the analysis of sensory properties, toxicity and storage properties of the processed food, as well as the extension of the treatment, are suggested as a conclusion of this research.

Lee et al. [53] applied DBD treatment at 250 W, 15 kHz and ambient air to cooked brown and white rice samples inoculated with *Escherichia coli* O157:H7 and *Bacillus cereus* for 5, 10 and 20 min. In the results, the DBD treatment reduced the number of pathogenic *E*. *coli* from 5.68 and 5.67 to 2.01 and 2.03 log CFU/g in the cooked brown and white rice, respectively, after 20 min of treatment. After treatment for 5, 10 and 20 min, the number of B. cereus in the cooked brown and white rice decreased significantly from 5.30 to 3.39 and from 5.29 to 0.05 log CFU/g, respectively. The number of total aerobic bacteria and coliform bacteria in cooked brown and white rice significantly decreased by about 2 log CFU/g after cold plasma treatment. Protein structure of pathogenic microorganisms were altered due to charged particles on the surface. As a result, pores were created in the pathogen membrane and enzyme activity was inhibited. Furthermore, reactive species cause oxidative damage to DNA and lipid peroxidation at the plasma membrane. Quality parameters such as as texture and lipid oxidation were affected.

Los et al. [54] inoculated *Bacillus atrophaeus*, (vegetative cells and endospores), *Aspergillus niger*, *Penicillium citrinum*, *Penicillium verrucosum*, *A*. *flavus/oryzae*, *A*. *candidus* and *P*. *chrysogenum* in organic wheat grains (*Triticum aestivum L*.). These samples were submitted to direct and indirect DBD for 5 to 20 min, after which they were stored closed at 15 °C for 24 h post-treatment retention time (PTRT). The treatment significantly reduced the level of all microorganisms adhering to the surface of wheat kernels. Both type of microorganism and treatment parameter exerted a great influence in the inactivation. For example, the mode of exposure did not significantly influence treatment efficacy in the case of *P*. *verrucosum* and *P*. *citrinum*. However, the highest inactivation levels for *A*. *niger*, *P. chrysogenum*, *A*. *flavus* and *B*. *atrophaeus* endospores and vegetative cells were obtained after 20 min of direct treatment compared to the indirect mode. Extending the direct treatment time from 5 to 20 min did not influence the inactivation effects of plasma against *A*. *candidus*, whereas a longer duration of direct plasma treatment significantly affected all other microorganisms tested.

**Table 4 nutrients-14-04653-t004:** Cereals treated with DBD at atmospheric pressure and ambient air.

Food	Microorganism	Treatment	Parameters	Log Reduction	Reference
Wheat and Barley	Microflora (bacteria and fungi) and inoculated bacteria (*E*. *coli*, *B*. *atrophaeus vergetativas*, *B*. *atrophaeus endospores* y *P verrucosum spores*).	80 kV 5–20 min.	Microbial control and seed germination.	Barley: 2.4 and 2.1 log CFU/g for bacteria and fungi respectively Wheat: 1.5 and 2.5 log CFU/g for bacteria and fungi respectively.	[55]
Cereal	*Bacilo* spp., *Lactobacillus* spp. and *E*. *coli* and *B*. *atrophaeus endospores*.	120 kV 5, 20, 30 min.	Bacterial inactivation, biofilms and spores.	5 min: reduced biofilms of *E*. *coli* spp., *B*. *subtilis* y *Lactobacillus* spp. > 3 log. 20 min: reduced B. atrophaeus spores in liquids by >5 log. 30 min: reduced spores on hydrophobic surface by > 6 and maximum reductions of 4.4 log were achieved with spores attached to the hydrophilic surface.	[51]
Brown rice	*Bacillus cereus*, *Bacillus subtilis*, *and Escherichia coli O157:H7*.	250 W, 15 kHz 5,10, 20 min.	Bacterial inactivation.	20 min: 2,30 log UFC/g.	[56]
Korean rice cakes	*Salmonella*, indigenous mesophilic aerobic bacteria.	25 kV 1 and 3 min.	Bacterial inactivation.	2 ± 0.1 log CFU/g in indigenous bacteria, inactivated indigenous yeasts and moulds and Salmonella by 1.7 ± 0.1 log CFU/g and 3.9 ± 0.3 log CFU/g, respectively.	[52]
Cooked white and brown rice	*Bacillus cereus (KCTC 3624)*, *E*. *coli O157:H7 (KCCM 40406*) and aerobic mesophilic and coliform bacteria.	250 W 5, 10,20 min.	Microbial analysis, physicochemical properties (pH, sticking, texture, sugar reduction, lipid oxidation).	*E*. *coli* inoculated on cooked brown and white rice at: 5.79 and 5.80 log CFU/g respectively. After treatment: 2.01 and 2.03 log CFU/g in brown and white cooked rice respectively. *Bacillus cereus* initial in brown and white rice at levels of 5.68 and 5.67 log CFU/g, respectively After the 5, 10 and 20 min treatment, B. cereus on cooked brown and white rice decreased from 5.30 to 3.39 and from 5.29 to 0.05 log CFU/g, respectively. Aerobic mesophilic bacteria and coliforms were reduced by approx. 2 log post treatment.	[53]
Organic wheat grains *(Triticum aestivum* L.*)*.	*Bacillus atrophaeus var*. *niger ATCC 937*, *Aspergillus niger ATCC 16404*, *Penicillium citrinum DSM 1179* and *Penicillium verrucosum DSM 12639*, *A*. *flavus/oryzae*, *A*. *candidus* and *P*. *chrysogenum*, spoilage flora.	80 kV 5 and 20 min.	Bacterial inactivation.	Higher levels of inactivation for *A*. *niger*, *P*. *chrysogenum*, *A*. *flavus* and endospores and vegetative cells of *B*. *atrophaeus* after 20 min direct mode of treatment.	[54]

## 5. Hybrid Technologies

A hurdle technology is a minimal processing technology that exploits synergistic between preservation treatments. The impact on food quality (sensory and nutritive properties) is minimized [57].

One of the most studied methods has been ultrasound-assisted plasma system. Chen et al. [58] proposed this technology to inactivate *E*. *coli* and yeast (*Saccharomyces cerevisiae*). After 30 min of treatment, the inactivation achieved was 6 log reductions with ultrasound assisted plasma, while with plasma system (without aeration condition) the inactivation achieved was 2 log reductions.

Similarly, Pan et al. [59] combined DBD technologies and ultrasound, inactivating *L*. *monocytogenes* under different growth temperatures (10, 25, 37 and 42 °C). They compared DBD treatment alone for different durations (0, 1, 2, 3 and 4 min) with the combination of ultrasound pretreatments for 0, 5, 10 and 15 min followed by plasma treatment for 2 min. The DBD effectiveness was increased due to the weakening effect on the membrane caused by ultrasound pretreatments. Based on these findings, growth temperature-mediated alterations in fatty acid profile are correlated with intracellular ROS levels and inactivation efficiency.

Umair et al. [60] also combined the effect of ultrasonication and DBD at 70 kV for 4 min in carrot juice. Total plate count was found after DBD treatment, ultrasonication and combined treatment 2.5, 3 and 3 log CFU/mL.

Metha and Yadav [61] determined the hurdle effect of combining atmospheric cold plasma and hydrothermal treatment on ascorbic acid, individual polyphenolic compounds, total phenolic content, and microbial inactivation. They applied DBD treatment at 60 kV for 10 and 15 min. Combined treatment of DBD followed by hydrothermal decreased the microbial growth.

An organic acid, an essential oil, an antimicrobial peptide, and a polymeric biocide are some of the most commonly used chemical antimicrobials. The combination of chemical antimicrobials and DBD has been proposed recently. Biocontrol agents such as bacteriophage with cold plasma have been used to improve microbicidal activity and ensure food safety [57]. Shan et al. [62] studied different concentrations of lactocin with DBD at 56 kV during different timings between 30 and 180 s on *Morganella* sp. The most effective treatment was found to be the synergic treatment of lactocin with 0.3 mg/mL with 120 s of DBD.

De la Ossa et al. [63] studied the antimicrobial effects of DBD with olive leaf extract (OLE) against *Escherichia coli*, *Staphylococcus aureus*, and *L*. *innocua*. In comparison to individual treatments, these combined treatments showed considerable antimicrobial activity.

Trevisani et al. [64] investigated the synergistic effect of DBD, Sodium Dodecyl Sulphate (SDS), and Lactic Acid (LA) on *L*. *monocytogenes* and verotoxin-producing *E*. *coli* in red chicory. Samples were pre-treated with SDS, or SDS + LA for 5, 10 or 15 min. After pre-treatment they were then submerged in deionised water and exposed to DBD at 19.15 V for 15 min. Both technologies reduced below the limit of quantification verotoxin-producing *E*. *coli* in red chicory. *L*. *monocytogenes* showed a higher tolerance to this sanitizing treatment and the level of inactivation was higher than 3 logs (3.77 Log CFU/cm^2^ vs control) only by increasing the duration of the washing step in LA+SDS to 15 min. This study demonstrated that Lactic Acid (LA), SDS, and atmospheric cold plasma (ACP) had synergistic effects on verotoxin-producing *E*. *coli* and *L*. *monocytogenes* inactivation.

Many researchers have claimed that plasma activated water’s antibacterial activity contributed to the synergistic effects through its physicochemical characteristics, including ROS, RNS, pH, and UV radiation [65].

Another approach is the use of a magnetic field with cold plasma to increase the density of the plasma by applying 0.587 T magnetic field to a plasma jet, the concentration and electron density of the cold plasma grew by 2.4 and 1.5 times, respectively [65].

## 6. Scale up Challenges and Industrial Application

Product properties such as surface roughness, moisture content and chemistry conditioned process efficacy [66]. Plasma inactivation efficiency is influenced by technological parameters, such as input power, gas composition and treatment time. Microorganism resistance to plasma treatment will also be affected by food intrinsic properties [67]. A plasma treatment’s efficacy can be affected by osmolarity and pH, while plasma reactive species may be diminished by lipid and protein content and antioxidant status [68]. An example of this was reported in [67]: *S*. *Enteritidis* cells were more susceptible treated by plasma at pH 5 as compared to pH 7. *S*. *typhimurium* and *Listeria monocytogenes* cells were more resistant towards plasma treatment at salt concentration 0% (*w*/*v*) compared to 6% (*w*/*v*).

Several publications have reported cytotoxic effects of plasma-activated water and plasma-activated medium (PAM) on eukaryotic cells [37]. Safety considerations must be taken into account when using plasma in food production and food processing. In plasma activated liquids, hydrogen peroxide has been implicated as a key cytotoxic species, but it is not the only toxic agent. Controlling and validating processes are essential to the safety of high-risk products such as food. Non-thermal treatments are more challenging to validate than heat treatments. In order to identify minimum concentrations and exposure times at which toxic or mutagenic effects can occur, and to define safe doses according to food targets, it is necessary to determine the minimum concentrations and exposure times that can occur [66].

In relation to direct versus indirect application of DBD treatment, Los et al. [54] found that the mode of exposure did not significantly influence the efficacy of treatment in the case of *P*. *verrucosum* and *P*. *citrinum* when longer treatment (20 min) was applied. However, higher inactivation levels for *A*. *niger*, *P*. *chrysogenum*, *A*. *flavus* and endospores and vegetative cells of *B*. *atrophaeus* were obtained after 20 min of direct mode of treatment as compared to indirect mode.

Surfaces of food packaging and contact surfaces that do not directly contact food tend to be relatively uniform and smooth, making them easier to treat and more acceptable to regulators [66].

The regulatory ramifications of developing technology differ globally. The United States Department of Agriculture (USDA), the Food and Drug Administration (FDA), and the Environmental Protection Agency (EPA) have all approved the use of cold plasma in food and food packaging [65].

The cold plasma source needs to be created, developed, and installed for industrial applications in a way that does not disrupt the ongoing production process. For laboratory operations, a smaller sample quantity or volume is necessary. The processing requirements are substantial. As a result, substantial volumetric scale-up is necessary for continuous and smooth operation. Additionally, maintaining plasma homogeneity during scale-up demands for creation of a suitable plasma source that can meet the plant’s requisite capacity [65]. Food applications have drawn more attention to atmospheric-pressure plasma generators but scaling discharges up to larger regions without compromising the plasma homogeneity is still a significant difficulty [66].

Food components may undergo a variety of chemical changes as a result of plasma exposure, such as the oxidation of sugars to organic acids, alteration of the amino acid residues in proteins, and peroxidation of lipids, which may produce hazardous metabolites including short chain aldehydes. Long-term exposure of cell lines to complex plasma-treated protein models has been shown to have cytotoxic and mutagenic effects [37]. However, other studies did not detect the mutagenic potential of plasma-treated media [66]. Identifying the differences in plasma devices, treatment protocols, and target compositions will be crucial to determine safe doses based on the application or food target in issue as well as lowest concentrations and exposure times at which an enhanced rate of harmful or mutagenesis effects can occur. Studies comparing the cytotoxicity and mutagenicity of cold plasma to those of the already recognised sterilants, sanitizers, and disinfectants are particularly necessary. Another obstacle addressed by the discipline of “plasma medicine” is the need for a measurable or adjustable plasma dosage for dietary substances. It is difficult to choose one or a few characteristics to define a plasma treatment dose due to the range of sources and species used in the produced effects. Given their wide range of products, the option of measuring the dose absorbed by the target, which is appropriate for medical applications, is incompatible with food analysis [66].

## 7. Strengths and Limitations

DBD cold plasma can be a promising technology to reduce microbial contamination. Consumers are increasingly demanding non-thermal technology to retain natural and fresh-like properties of foods.

However, DBD can impair sensorial and lipid oxidation parameters in some kinds of food. It is important to establish process conditions such as input power and treatment time depending on foodstuff.

For the industrial application of DBD cold plasma treatment, a number of challenges have to be overcome. Firstly, all the variables that can affect the efficacy of the treatment have to be managed. Among them is the direct versus indirect application of plasma, which creates a challenge for its regulation. For the validation of the process, the cytotoxic and mutagenic activity of the treatment found in some works related to cold plasma must be clarified. The lack of uniformity in reporting is another major challenge in terms of treatment dosimetry.

## 8. Conclusions

According to the results obtained in this literature review, cold plasma treatment of food is effective in inactivating microbes. In all the studies reviewed, a significant logarithmic reduction of inoculated bacteria and/or altering flora of the food in question was achieved. In the experiments carried out with meats and fish, the general results show alteration of the qualitative characteristics of the treated product, such as taste and colour. In this group of foods, the food matrices exert a protective effect towards microorganisms, as they have high concentrations of organic compounds. Another important fact in this group of experiments is that the type of reactive species and their concentrations depend directly on the working gas used. Therefore, the composition of the gas will determine the microbial inactivation rates. DBD plasma applied to meats and fish is generally considered to be an effective antimicrobial treatment and thus, also prolongs the shelf life of the treated food. However, it must be improved in terms of lipid oxidation and other quality characteristics that are impaired by the application of the process.

In the fruits and vegetables studied, DBD cold plasma exerted its antimicrobial effect and also prolonged their shelf life. In this food group, research on optimal packaging conditions has been carried out due to differences in electrical conductivity materials. It is of great importance to select the right DBD conditions (power and time) for each food to control the type and concentration of reactive species, and on the other hand to avoid surface and nutritional damage to the food. A well-studied factor in fruit and vegetables are the particularities of the food surface, as these can provide niches for bacteria, creating a physiological barrier against DBD.

In the dairy group, it is concluded that the efficacy of the treatment is affected by the type of microorganism. Similarly to fruits, attention should be paid to the surface of the cheese to ensure its effectiveness. In this food group the changes in quality characteristics due to the treatment were acceptable.

In the cereal group, DBD treatment improves the microbial quality of the food, the process should be optimized to reduce the undesirable quality and physicochemical modifications.

The combination of DBD with other non-thermal technologies such as ultrasonics, with thermal or chemicals compounds showed promising results. The combined effect is much more significant in improving both safety and the quality of products. Hurdle technologies, including DBD, have very strong potential for food application.

## Figures and Tables

**Figure 1 nutrients-14-04653-f001:**
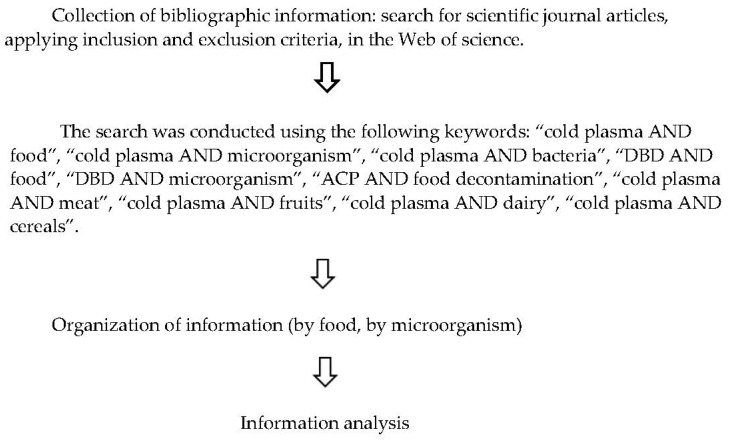
Flow diagram of review.

**Figure 2 nutrients-14-04653-f002:**
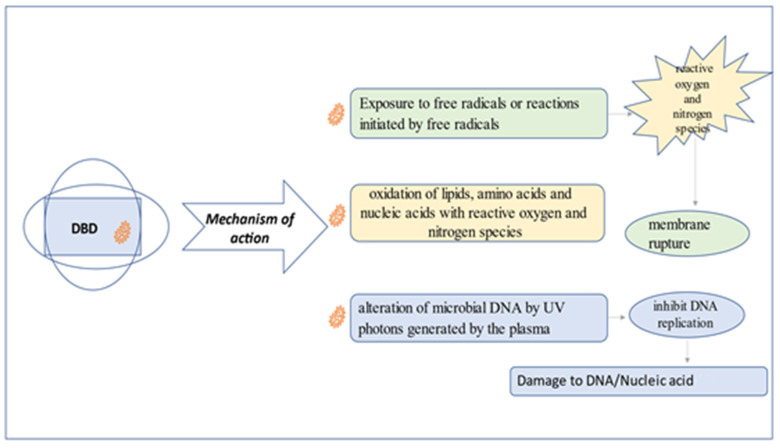
DBD mechanism of action in food for decontamination.

**Table 1 nutrients-14-04653-t001:** Meat and fish treated with DBD cold plasma at atmospheric pressure and ambient air.

Food	Microorganism	Treatment	Parameters	Log Reduction	Citation
Poultry meat	Native mesophilic aerobic bacteria, *Salmonella* and Tulane virus.	24 kV and 3 min.	Microbial inactivation, nitrates and nitrites, oral toxicity and storage quality.	0.70 ± 0.12 log CFU/cube, 1.45 ± 0.05 log CFU/cube, 1.08 ± 0.15 log CFU/cube, respectively.	[21]
Poultry meat	Psychrophilic, *Campylobacter jejuni*, *Salmonella typhimurium*.	60, 180, 300 s.	Microbial analysis, ozone formation, colour.	60 s: 0.5, 0.7 and 0.4 red. Log. Respectively. 180 s: reduced psychrophils by an additional 0.6 logs.	[22]
Lamb meat	*Brochothrix thermosphacta*.	80 Kv, 30 s and 5 min.	Microbial reduction.	30 s: below detection levels. 5 min: 2 log cycles (complexed meat model medium and adherent cells).	[23]
Meat	*Listeria monocytogenes*, *Staphilococcus aureus* and *Escherichia coli (O157:H7)*.	60,70 y 80 kV 60 s.	Bacterial inactivation.	60 s: undetectable in PBS.	[24]
Pork meat	Altering microflora.	85 kV 60 s.	Bacterial inactivation.	Significant logarithmic reduction in treated group.	[25]
Fresh mackerel *(Scomber scombrus)*	Spoilage bacteria (total aerobic psychrotrophic, pseudomonads and lactic acid bacteria).	70 y 80 kV 1, 3 and 5 min.	Microbial parameters and quality (pH, colour, lipid oxidation, protein structure, water distribution).	Psychrotrophic bacteria, LAB and pseudomonas were significantly reduced (*p* < 0.05). Higher inactivation at higher time and voltage.	[26]
Bolti fish (*tilapia nilótica*)	Altering microflora (TVC or variable total count).	40, 50 y 60 kV 1, 2, 3 and 4 min.	Endogenous enzyme activity, quality characteristics and quality during storage.	60 kV and 4 min, increased shelf life of tilapia fish to 10 days.	[27]
Herring *(clupea harengus)*	Total aerobic mesophiles, psychrotrophs, pseudomonads, lactic acid bacteria and *enterobacteriae*.	70, 80 kV 5 min.	Microbiological analysis, pH, barbituric acid reactive substances (TBARS), colour, water mobility during storage.	Initial LAB counts: 2.10 ±0.01 (80 kV for 5 min) at 3.80 ± 0.71 CFU/g (Control) on day 1. Day 11: 5.10 ± 0.02 CFU/g (80 kV for 5 min) at 6.05 ± 0.07.	[28]
Mackerel (*Scomber japonicus*)	Altering bacterial flora (TVC or variable total count).	10, 20, 30, 40, 50, 60 and 70 kV. 15, 30, 45, 60 and 75 s.		45 s 60 kV: TVC decreased from 5.02 ± 0.48 log CFU/g fish flesh to 2.64 ± 0.16 log CFU/g fish flesh. 60 s: 3.15 log cycles 75 s: no longer significantly reduced.	[29]
Dry blackmouth triggerfish	*Staphylococcus aureus and Bacillus cereus*.	15, 10, 20 and 30 min.	Physicochemical properties (pH colour, reactive barbituric acid) and sensory quality.	*S. aureus* and *B. cereus*:15, 10, 20 and 30 min: 0.10 and 1.03 log CFU/g and 0.14 to 1.06 log CFU/g, respectively. 30: >90% reduction of *S. aureus* and *B. cereus* with no overall adverse changes.	[30]

**Table 2 nutrients-14-04653-t002:** Fruits and vegetables treated with DBD at atmospheric pressure and air.

Food	Microorganisms	Treatment	Parameters	Log Reduction	Citation
Strawberries	Microflora (mesophilic bacteria, yeasts and moulds).	60 kV 5 min.	Ozone concentration, microbial inactivation, colour, firmness and respiration rate.	2 log within 24 h post treatment.	[15]
Melon	Total mesophilic and psychrotrophic bacteria, *lactococci*, *lactobacilli* and yeasts.	15 kV, 30 min and 60 min.	Qualitative (titratable acidity, soluble solid content, dry matrix, colour, texture) and microbiological characteristics.	Mesophilic and lactic bacteria: 60 min: 3.4 and 2 Log CFU/g respectively. significant increase in shelf life.	[31]
Red chicory	*Escherichia coli, Listeria monocytogenes.*	15 kV, 15, 30 min.	Bacterial reduction, antioxidant activity and quality.	*E. coli* 15 min: 1.35 log MPN/cm^2^ *L. monocytogenes* 30 min: 2.2 log UFC/cm^2^.	[32]
Carrot	Natural carrot microflora (mesophiles, yeasts and moulds).	60, 80 and 100 kV 5 min.	Inactivation of spoilage bacteria.	Maximum reduction of 2,1 log 10 CFU/g at 100 kV.	[33]
Packaged vegetable mix (grape tomato, romaine lettuce, red cabbage, carrot))	*Salmonella,* mesophilic aerobic bacteria, yeast and moulds.	hasta 50 kV 3 min.	Bacterial reduction.	Polyethylene container Cherry Tomatoes: 1.2 log CFU/tomato Polyethylene terephthalate packaging achieved 0.8 log CFU/tomato. *Salmonella*, yeast and mould on romaine lettuce, red cabbage and carrot remained ~1 log CFU/sample.	[34]
Strawberries and spinach	*E. coli y L. innocua.*	2.5 min.	Shelf life, microbiological activity, quality: colour, pH, firmness, total soluble solids (Brix).	*Static mode: Strawberries(*E. coli*): 2.0 log 10 CFU Spinach(*E. coli*):2.2 log CFU Strawberries (*L.innocua*): 1.3 log CFU Spinach (*L.innocua*):1.7 log CFU *Continuous mode: L. innocua:0.9 and 2.9 for strawberries and spinach respectively.	[35]
Strawberries	Microflora	60 kV 10, 15 and 30 min.	Bacterial inactivation, phenolic compounds, antioxidant capacity, physico-chemical parameters.	At 60 kV: 2 log extended the shelf life of the strawberry for 3 days at 25 °C.	[36]
Cherry tomato and strawberry	*Escherichia coli, Salmonella enterica* typhimurium and *Listeria monocytogenes.*	70 kV 10–120 s and 300 s.	Microbial inactivation	Tomatoes, 10 s: *Salmonella* undetectable 60 s: *L. monocytogenes* at 5.1 log CFU/sample and *E. coli* at undetectable levels. 120 s: *L. monocytogenes* to undetectable levels. 300 s: strawberries: reductions of *E. coli*, *Salmonella* and *L. monocytogenes* 3.5, 3.8 and 4.2 log CFU/sample respectively.	[37]
Cherry	Non-specific bacterial colonies.	40–80 kV 60–140 s.	Microbial inactivation and quality (spoilage rate, respiration rate, total soluble solids, total phenolic compounds, flavonoids, anthocyanin, VC, titratable acidity).	At 60 kV in 60 s it can inactivate most microorganisms.	[38]
Romaine lettuce in package	*Escherichia coli O157: H7* and mesophilic aerobic microorganisms, yeasts and moulds.	34.8 kV 5 min	Microbial inactivation, colouring, CO_2_ generation, weight loss, surface morphology of lettuce.	*E. coli O157: H7,* mesophilic aerobic and yeasts and moulds during storage: 0.8 to 1.5, 0.7 to 1.9 and 0.9 to 1.7 log CFU/g respectively.	[39]
Romaine lettuce	*Escherichia coli O157:H7*, *Salmonella, Listeria monocytogenes* and *Tulane virus.*	34.8 kV 5 min.	Microbial inactivation.	*E. coli O157:H7, Salmonella, L. monocytogenes*, and Tulane virus was 1.1 ± 0.4, 0.4 ± 03, 1.0 ± 0.5 log CFU/g, and 13 ± 0.1 log CFU/g, respectively.	[40]
Pears	Aerobic mesophilic bacteria, yeast and mould.	65 kV 1 min.	Bacterial growth, organoleptic properties and quality, pectin activity, methylesterase.	At 65 kV for 1 min, quality attributes were preserved, inhibiting the microorganisms studied.	[41]
Arugula in package	Total viable flora, pseudomonas spp., yeasts and moulds and lactic acid bacteria.	6 kV 5–20 min.	Microbiological analysis, quality, (texture, pH, colour).	10 min: where the microbial load decreased by 1.020, 0.298, 0.493 and 0.996 log CFU/g for total viable flora, Pseudomonas spp., yeasts/moulds and LAB, respectively.	[42]
Grape tomato	*Salmonella.*	35 kV 3 min.	Quality (colour, firmness, weight loss, lycopene and ascorbic acid concentration, microbial safety).	Immediate effect: 3.3 ± 0.5 log CFU/tomato.	[43]
Apple	*Salmonella (Salmonella Typhimurium, ATCC 13311; Salmonella Choleraesuis, ATCC 10708)* and *E. coli. (ATCC 25922, ATCC 11775).*	200 W 30, 60, 120, 180 and 240 s.	Bacterial inactivation.	Ranged from 1.3 to 5.3 and 0.6 to 5.5 log CFU/cm^2^ for *Salmonella* and *E. coli*, respectively. Both bacteria decreased significantly at 180 and 240 s.	[44]
Blueberries	Native bacteria and fungi (yeast and mould).	36 V 0, 2, 4, 6, 8 and 10 min.	Quality, in terms of microbial growth, antioxidant value and decay rate.	10 min: total bacterial count showed a decrease of 2.01 log CFU g-1FW. FW and fungal counts decreased by 0.58 log CFU g^−1^FW.	[45]
Strawberry	Aerobic mesophilic bacteria and yeasts/moulds.	30 W y 12 kV 5–20 min.	Decontamination and quality (weight loss, colour changes, firmness).	Bacterial and yeast/mould reduction after 20 min reached 1.46 and 2.75 log CFU/g.	[46]
Fresh vegetables, fruits and nuts.	*Escherichia coli O157:H7, Salmonella Typhimurium*, and *Listeria monocytogenes*,	51,7 W 20 min.	Antimicrobial effect of atmospheric cold plasma.	Various levels of efficacy were determined, depending on food type and surface area. Effective for vegetables and fruits.	[47]

**Table 3 nutrients-14-04653-t003:** Dairy products treated with DBD cold plasma at atmospheric pressure and ambient air.

Food	Microorganism	Treatment	Parameters	Log Reduction	Reference
Fresh cheese and model cheese	*Listeria innocua*.	100 kV 5 min.	Bacterial reduction.	Direct exposure: 1.6 and 3.5 log CFU/g, in fresh cheese and model cheese respectively. Indirect exposure: 0.8 and 2.2 log CFU/g respectively.	[49]
Cheese	*Escherichia coli*, *Listeria innocua*.	30, 50, 70 W 0, 1, 3, 5, 7 m.	Microbial inactivation.	50 W for 10 min: 4.75 ± 0.02 and 0.72 ± 0.01, respectively.	[50]

## Data Availability

Not applicable.

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
