# Peer review of "Dielectric Barrier Discharge for Solid Food Applications"

_nutrients, 2022, doi:10.3390/nu14214653_

Round 1
Reviewer 1 Report
The review presents an interesting contribution to the field of food and nutrition research, but I found it missing key points. First of all, please complete the review with the purpose and methodologies. Every review should include an objective ("the purpose of the review conducted was to search the literature on...") and the research questions it answers. In addition, the methodology is missing.... How were the sources selected? In what database? What phrases/tags were used? Which sources were rejected and which were accepted (eligibility criteria)? Why? How many sources were finally qualified? How many helped to write the introduction, and how many gave answers to the researcher's questions? I recommend the PRISMA Scope methodology for conducting reviews. In addition, the methodology should include a graph showing the selection of sources and the basic information I mentioned above. At the end of the review, please include a section with its strengths and limitations.
Author Response
Dear Editor and Reviewers,
We are sending to you all the changes suggested and the answers to the questions about the review: “Dielectric Barrier Discharge for solid food applications” (by M.F. Figueroa-Pinochet, M.J. Castro-Alija, B.K. Tiwari, J.M. Jiménez, M. López-Vallecillo, M.J. Cao & I. Albertos) that needed to be clarified. We hope these comments are useful in answering to all the questions raised.
Reviewer 1
Comments to Author
Reviewer’s comment:
The review presents an interesting contribution to the field of food and nutrition research, but I found it missing key points. First of all, please complete the review with the purpose and methodologies. Every review should include an objective ("the purpose of the review conducted was to search the literature on...") and the research questions it answers.
Authors’ response:
The authors agree with the reviewer and the objective has been added at the end of the introduction.
“The purpose of this review conducted was to provide the potential of the application of DBD in different solid food groups. It included the mechanism of action in microbial inactivation. The combination of DBD with other technologies has been discussed (hybrid technologies and hurdle technologies). A hurdle technology is a minimal processing technology that exploits synergistic interactions between DBD and other preservation treatment. It concluded with a discussion of the main challenges and industrial application”.
Reviewer’s comment:
In addition, the methodology is missing.... How were the sources selected? In what database? What phrases/tags were used? Which sources were rejected and which were accepted (eligibility criteria)? Why? How many sources were finally qualified? How many helped to write the introduction, and how many gave answers to the researcher's questions? I recommend the PRISMA Scope methodology for conducting reviews. In addition, the methodology should include a graph showing the selection of sources and the basic information I mentioned above.
Authors’ response:
The authors agree with reviewer’s comment. The methodology section has been added (2. Methods).
Reviewer’s comment:
Please include a section with its strengths and limitations.
Authors’ response:
The strengths and limitation section has been added in this regard (7. Strengths and limitations).

Reviewer 2 Report
The authors present an interesting review on the applications of DBD plasmas in food processing. With their manuscript, they extend other authors' more general overview articles of plasma food processing by providing additional details on the narrowed-down subfield. Particularly the focus on DBD plasmas and solid foods provides a worthwhile extention over existing reviews, with further information taken from very recently published articles that previous state of the art reviews did not yet contain. The structure of the manuscript is useful, clear and intuitive. Moreover, mechanisms are clearly introduced in sufficient detail. Occasional references to other, similar DBD applications in related (eg health) fields could be used more extensively, but are provided sufficiently well for the main intended readership.
That being said, there is a number of minor points, which the authors are strongly encouraged to rephrase in order to avoid possible misunderstandings among the readers.
1. Lines 11 & 39. The use of non-thermal plasmas for food processing has more widely entered this specific field relatively recently. As such, it is a quickly developing branch of plasma research. However, atmospheric-pressure non-thermal plasmas themselves are far from being a novel technology, but have been used in technical applications well over 150 years. The authors should phrase their text (abstract and introduction) accordingly and avoid shortened, absolute statements about the novelty of the technology.
2. Lines 39-45. There is some confusion in existing literature about physical definitions, which the authors unfortunately mirror in their manuscript, henceforth spreading the confusion and misunderstanding among users of the technology. Shortly said, state of matter refers to equilibrium states, whereas non-thermal plasmas are highly non-equilibrium systems. Hence, the term 4th state of matter correctly describes thermal plasmas. Contrary to that, DBDs and other non-thermal plasmas are correctly described as gases.
3. Line 65 & figure 1. The term bombardment by free radicals is misleading and should be avoided. Bombardment implies notable kinetic energy or gas temperature, which is not the case for mechanisms in DBDs. Better terms are e.g exposure to free radicals or reactions initiated by free radicals.
The authors should address these three issues prior to the publication of the manuscript.
Author Response
Dear Editor and Reviewers,
We are sending to you all the changes suggested and the answers to the questions about the review: “Dielectric Barrier Discharge for solid food applications” (by M.F. Figueroa-Pinochet, M.J. Castro-Alija, B.K. Tiwari, J.M. Jiménez, M. López-Vallecillo, M.J. Cao & I. Albertos) that needed to be clarified. We hope these comments are useful in answering to all the questions raised.
Reviewer 2
Comments to Author
Reviewer’s comment:
The authors present an interesting review on the applications of DBD plasmas in food processing. With their manuscript, they extend other authors' more general overview articles of plasma food processing by providing additional details on the narrowed-down subfield. Particularly the focus on DBD plasmas and solid foods provides a worthwhile extention over existing reviews, with further information taken from very recently published articles that previous state of the art reviews did not yet contain. The structure of the manuscript is useful, clear and intuitive. Moreover, mechanisms are clearly introduced in sufficient detail. Occasional references to other, similar DBD applications in related (eg health) fields could be used more extensively, but are provided sufficiently well for the main intended readership.
That being said, there is a number of minor points, which the authors are strongly encouraged to rephrase in order to avoid possible misunderstandings among the readers.
Reviewer’s comment:
Lines 11 & 39. The use of non-thermal plasmas for food processing has more widely entered this specific field relatively recently. As such, it is a quickly developing branch of plasma research. However, atmospheric-pressure non-thermal plasmas themselves are far from being a novel technology, but have been used in technical applications well over 150 years. The authors should phrase their text (abstract and introduction) accordingly and avoid shortened, absolute statements about the novelty of the technology.
Authors’ response:
The authors agree with reviewer’s comment. The abstract and introduction has been rewritten to avoid absolute statements about the novelty of the technology. The term novel has been removed. Novel technology has been replaced by “non-thermal technology” or “preservation technique”.
Reviewer’s comment:
Lines 39-45. There is some confusion in existing literature about physical definitions, which the authors unfortunately mirror in their manuscript, henceforth spreading the confusion and misunderstanding among users of the technology. Shortly said, state of matter refers to equilibrium states, whereas non-thermal plasmas are highly non-equilibrium systems. Hence, the term 4th state of matter correctly describes thermal plasmas. Contrary to that, DBDs and other non-thermal plasmas are correctly described as gases.
Authors’ response:
The authors agree with reviewer’s comment. This section has been rewritten: “Another non-thermal technology is cold plasma. Industrial cold plasma equipment has not been implemented in food industry yet. Cold plasma is composed by gas molecules, positive or negative ions, free radical… The gas temperature is close to ambient temperature and it is obtained at atmospheric or reduced pressure (vacuum). Cold plasma does not present a local thermodynamic equilibrium [9-10]”.
Reviewer´s comment:
Line 65 & figure 1. The term bombardment by free radicals is misleading and should be avoided. Bombardment implies notable kinetic energy or gas temperature, which is not the case for mechanisms in DBDs. Better terms are e.g exposure to free radicals or reactions initiated by free radicals.
Authors´ response:
The authors agree with the reviewer in the convenience of replacing the sentence in the manuscript and in the Figure 2.
In the manuscript, it has been replaced by: “Reactive oxygen and nitrogen species generated in DBD can lead can lead to cell membrane rupture, or to the oxidation of lipids, amino acids and nucleic acids with reactive oxygen and nitrogen species”. Regarding of Figure 2 has been modified by “exposure to free radicals or reactions initiated by free radicals” following to reviewer´s suggestion.

Round 2
Reviewer 1 Report
Current form of the manuscript is acceptable. Thank you for your revision. Great work.